# Development of a Detection System for Types of Weeds in Maize (*Zea mays* L.) under Greenhouse Conditions Using the YOLOv5 v7.0 Model

Oscar Leonardo García-Navarrete [1,2], Oscar Santamaria [3], Pablo Martín-Ramos [1], Miguel Ángel Valenzuela-Mahecha [2] and Luis Manuel Navas-Gracia [1,*]

1   TADRUS Research Group, Department of Agricultural and Forestry Engineering, Universidad de Valladolid, 34004 Palencia, Spain; oscarleonardo.garcia@uva.es (O.L.G.-N.); pmr@uva.es (P.M.-R.); luismanuel.navas@uva.es (L.M.N.-G.)
2   Department of Civil and Agricultural Engineering, Universidad Nacional de Colombia, Bogotá 111321, Colombia; olgarcian@unal.edu.co (O.L.G.-N.); mavalenzuelam@unal.edu.co (M.Á.V.-M.)
3   Department of Crop Science and Forestry Resources, Universidad de Valladolid, 34004 Palencia, Spain; oscar.santamaria@uva.es
*   Correspondence: luismanuel.navas@uva.es

**Abstract:** Corn (*Zea mays* L.) is one of the most important cereals worldwide. To maintain crop productivity, it is important to eliminate weeds that compete for nutrients and other resources. The eradication of these causes environmental problems through the use of agrochemicals. The implementation of technology to mitigate this impact is also a challenge. In this work, an artificial vision system was implemented based on the YOLOv5s (You Only Look Once) model, which uses a single convolutional neural network (CNN) that allows differentiating corn from four types of weeds, for which a mobile support structure was built to capture images. The performance of the trained model had a value of mAP@05 (mean Average Precision) at a threshold of 0.5 of 83.6%. A prediction accuracy of 97% and a mAP@05 of 97.5% were obtained for the maize class. For the weed classes, *Lolium perenne, Sonchus oleraceus, Solanum nigrum*, and *Poa annua* obtained an accuracy of 86%, 90%, 78%, and 74%, and a mAP@05 of 81.5%, 90.2%, 76.6% and 72.0%, respectively. The results are encouraging for the construction of a precision weeding system.

**Keywords:** deep-learning; precision agriculture; convolutional neural network (CNN); computer vision; precision weeding

## 1. Introduction

According to data from the Food and Agriculture Organization of the United Nations (FAO), maize (*Zea mays* L.) is the main cereal crop planted in the world, reaching a production of 1.12 billion tons in 2021 [1], so its integrated management with respect to weeds is essential to maintain productivity [2]. Weeds, as plants, are fast-growing and compete for space, water, nutrients, and sunlight. On the other hand, weeds negatively influence crop yield and quality [3]. According to [4], yield damage caused by weeds can reach 42% of agricultural production. Currently, there are several weeding methods, including chemical weeding using pre- and post-emergence herbicides, which are used daily as mitigation measures, but these generate high environmental impacts; physical weeding using plastic covers, but the inputs used increase the cost of production; mechanized weeding with electric or manual tools, but due to its high cost in terms of labor it is not an economically viable option, and they also require time and human effort that can cause health impacts [5]; and biological weeding through the use of microorganisms, among others [6]. Weed management, by any method, presents environmental problems or economic disadvantages, so finding a way to reduce these issues is a challenge. The implementation of new technologies, such as artificial intelligence combined with machine

vision applied to agriculture, is presented as an effective solution to address these challenges. The combination of these technologies allows us to locate and classify plants in various conditions, thus differentiating the crops from weeds [7]. To perform precision weeding, the first step is the accurate detection and identification of weeds [8]. However, in practice, weed detection faces several problems, such as similarities in colors, textures and shapes, as well as occlusion effects and variations in illumination environments. To cope with these difficulties, both traditional and deep learning-based machine vision offer effective solutions [9].

The combination of computer vision and deep learning techniques creates a powerful tool for object detection and classification in real time; one of them is YOLO (You Only Look Once), developed by [10] in 2015, where he proposes an object detection algorithm that consists of training a deep CNN (convolutional neuronal network) that predicts for each class and the precise position of objects in the images. The development of YOLO revolutionized object detection by conceptualizing it as a one-step regression problem, starting from image pixels and arriving at bounding boxes and class probabilities. Its "unified" approach allowed for the simultaneous prediction of multiple bounding boxes and class probabilities, improving both speed and accuracy [10,11]. Since the first version of YOLO in 2015, it has undergone a rapid evolution, becoming a family, with the latest version, YOLO-v8, being released in 2023. Although the original author, Joseph Redmon [10], ceased his contribution to the development of YOLO-v3 [12], several researchers have continued to improve the effectiveness and potential of the core "unified" approach. One of the most stable versions is YOLO-v5 [13], due to the fact that it was the first native version of YOLO architectures written in PyTorch rather than Darknet [14]. Although Darknet is considered a low-level flexible research framework, it was not specifically designed for production environments, and its employability in the scientific and development field was lower; on the contrary, PyTorch provided an established ecosystem with a broader subscriber base in the computer vision community, as well as offering a support infrastructure that facilitates deployment on mobile devices.

YOLO-v2 introduced the concept of "anchor box machine learning", utilizing k-means to select boxes resembling the dimensions of truth boxes in the training set. Five anchor boxes were initially chosen based on the COCO dataset, but limitations arose when applied to single datasets with significant differences. YOLO-v5 addressed this by integrating the anchor box selection into the architecture, allowing automatic learning for dataset-specific anchor boxes and streamlining training [15,16]. The YOLO architecture comprises three main components: a feature extraction backbone, a feature-aggregating neck, and a detection-generating head [16]. YOLO-v5, a PyTorch-native version, builds on YOLO-v4 contributions, refining computer vision techniques for enhanced performance [13,14]. The implementation of the different YOLO versions in agriculture has generated several applications; for example, in [17], the author proposes a modification to YOLO-v4 to improve efficiency and accuracy in the identification of weeds in a sesame crop, achieving a mAP (mean Average Precision) of 96.16%. In [18], the author proposed a lightweight weed detection model called YOLO-v4-weeds for weeds among carrot seedlings. Specifically, the backbone of the original YOLOv4 was replaced by MobileNetV3-Small, achieving a mAP of 88.46%. In the work of [19], they train versions of YOLO v3, v4, and v5 to detect and delimit the presence of insects in the field, finding that YOLO-v5 offers the best insect detection accuracy, with a (mAP) of 99.5%. In the research of [20], he evaluated four versions of YOLO in the detection of weeds in different turf settings, and his results showed that YOLO-v8l obtained the highest performance, with a mAP of 97.95%; however, the inference times in the YOLO-v8, YOLO-v7, and YOLO-v6 versions were around 30 ms, which is higher than YOLO-v5, which was less than 20 ms, concluding that YOLO-v5 is faster in new image inference times.

In the work of [21], he evaluated three deep learning-based methods, YOLO-v3, CenterNet, and Faster R-CNN, and the three methods obtained an average accuracy of 97%, but YOLO-v3 showed a mAP of 97.1%, with the highest of the three methods evaluated.

YOLO algorithms are implemented on different platforms for weed identification, as in [22], where unmanned ground vehicles (UGVs) were used for weed identification and removal in lettuce crops using the YOLO, Faster R-CNN, and SSD Mobile models. In [23], he used YOLO-v5 to build a real-time laser weeding robot in three crops: okra, bitter gourd, and sponge squash and four weed species, achieving a mAP of 88%. It has also been used in applications with UAVs (unmanned aerial vehicles); for example, [24] used a UAV to capture images and process them with YOLO-v7 in order to detect the weed Mercurialis annua in a sugar beet crop, obtaining a mAP of 62.1%. Similarly, [25] evaluated the performance of YOLO-v5 in the classification of crops and weeds with images taken with a UAV and found that the model obtained a accuracy of 65% mAP. In [26], the author evaluated three lightweight deep learning architectures, YOLOv4-tiny, YOLOv4-tiny-3lm and YOLOv7-tiny, for real-time weed detection in horticulture, with the results suggesting that the YOLOv4-tiny-3l model achieves the highest F1 score of 80.56%, reaching a mAP@0.5 of 83.38%. In the work of [27], the researchers built a field robot WeedDetwctRobot based on a modified version of YOLO-v5 called YOLO_CBAM for Solanum Rostratum detection, obtaining a mAP of 90.51% versus YOLO-v5, which obtained 86.14%.

In a corn crop, [28] used aerial images for weed detection using the versions of YOLO v4 and v5, obtaining a mAP of 73.1% with YOLO-v5s and 72.0% with YOLO-v4. In [29], he built a precision spraying robot for corn cultivation using the different versions of YOLO-v5, and as a result, he obtained a mAP of 89.4%; despite the fact that he did not obtain the best performance, his processing time was less than the other versions. In the work of [30], he built a robot for the elimination of weeds with a blue laser in corn seedlings, and he used the YOLOX version as the detection method, obtaining an average detection rate of 92.45% for corn and 88.94% for weeds. The use of YOLO algorithms in weed detection is a powerful tool that, with proper training and the necessary adjustments for each crop, will revolutionize weed control methods in the world. In this research, we present the results of the evaluation of the performance of the YOLO-v5s v7.0 model in the development of a detection system for four types of weeds in the cultivation of corn (*Zea mays* L.) under greenhouse conditions.

## 2. Materials and Methods

### 2.1. Materials

2.1.1. Biological Material

Sweetcorn (*Zea mays L.*) seeds were used. The seeds were sown in a raised bed containing substrate and distributed in forty-six containers; three seeds were placed per container, obtaining 138 corn plants, and the experimental period lasted 60 days, accounting for the entire vegetative period of corn development. It is considered that the phenological stage of vegetative development is the most sensitive for the crop due to the requirement of nutrients and its competition with weeds [2,31]. Four types of weeds (*Lolium perenne, Sonchus oleraceus, Solanum nigrum,* and *Poa annua*) were used; the weeds were sown randomly in two crop stages for a total of 966 weed plants during the whole experiment. The stages are described below:

— Stage 1: Considered to range from day 1 of sowing until day 31. In this stage, the weeds were randomly planted with a distribution of 6 weeds around each corn plant, obtaining 12 weed specimens per container for a total of 552. On day 31, the weeds were eradicated due to their size, considering that weeding is carried out 30 days after planting for commercial crops.

— Stage 2: ranged from day 32 days after sowing until day 60 of cultivation. After eradication, the weeds were randomly sown again with a distribution of 9 plants per container, simulating a regrowth of the same, obtaining a total of 414 in this stage.

The crop was located in the air-conditioned greenhouse of the University of Valladolid, La Yutera campus. The experiment was carried out from 2 February to 4 April 2023. The greenhouse maintained temperature set point values at 24 °C and 60% relative humidity, with automatic irrigation provided during the experiment.

2.1.2. Image Acquisition System

To capture the images, a mobile aluminum support structure was built on the raised bed of the greenhouse. The structure is on guide wheels that allow controlled displacement, as shown in Figure 1. The vision system consists of a Canon EOS 850D digital camera (Ohta-ku, Canon Inc., Tokyo, Japan), with a 22.3 × 14.9 mm² CMOS sensor located one meter above the substrate of the crop and an EF-S 15–55 mm f/4–536 IS STM lens (Tanzi, Canon Inc., Taichung, Taiwán) with a PL-C 58 mm circular polarizing filter was used. The lighting system was built with two 6000 K white light LED lamps configured in a matrix of 6 × 6 LEDs. Each lamp was fitted with a ROSCO polarizing filter with a transmission of 38% (1.5 f/stop) to avoid glare on the plant leaves, and the scene was additionally isolated from outside light with white polypropylene panels.

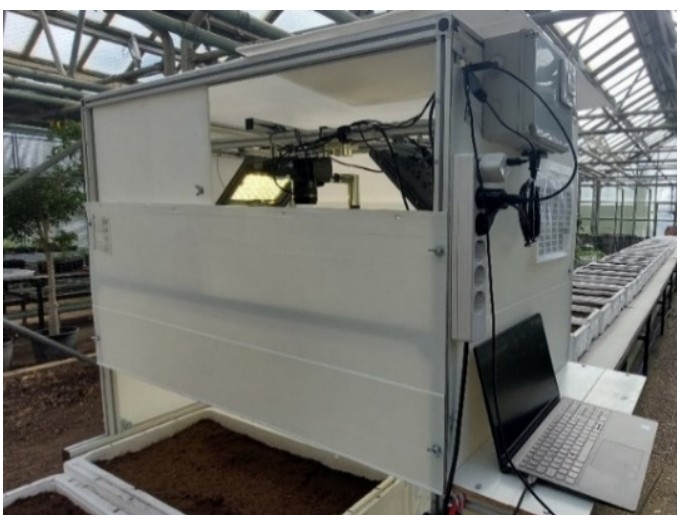

**Figure 1.** Mobile image acquisition structure.

*2.2. Methodology*

2.2.1. Images Capture

Images were taken in RGB with a resolution of 6000 × 4000 pixels for 35 days, distributed from the moment of plant emergence (day 12) to the end of the phenological stage of vegetative development (day 60), obtaining a total of 1640 images.

2.2.2. Image Selection and Labeling

The images suitable for training were selected, eliminating those that were out of focus or where the plant was so large that it went out of the scene; these were found in the last days of the vegetative stage. In the selection, 240 images were eliminated, leaving a total of 1400 images. For the labeling of the images, the open-source tool LabelImg was used, which allows users to save the labels in the YOLO format. Five label classes were used according to the species used, abbreviating their names for better handling, as shown in Figure 2. The label "maize" was used for *Zea mays* L., "mh1" for *Lolium perenne*, "mh2" for Sonchus oleraceus, "mh3" for *Solanum nigrum*, "mh4" for *Poa annua*, and "mh4" for *Poa annua*.

A set of 4200 maize bounding box labels and 10,322 weed labels were obtained. For training, the set of 1400 images was randomly divided into three: 70% for training with 980 images, 25% for validation with 350 images, and 5% for testing with 110 images.

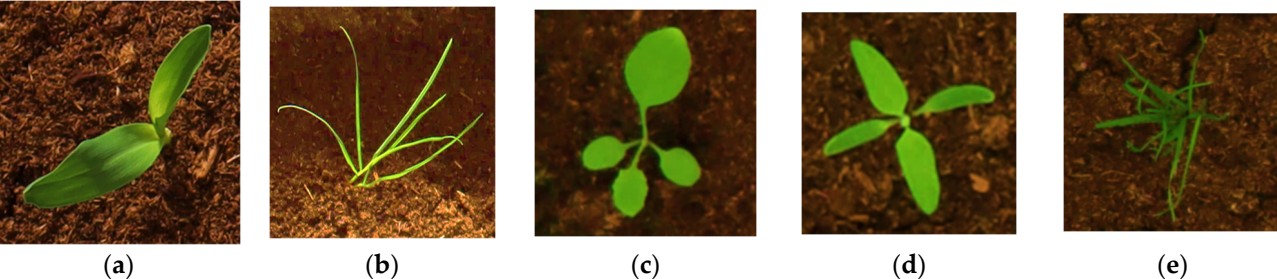

(**a**)      (**b**)      (**c**)      (**d**)      (**e**)

**Figure 2.** (**a**) Maize class: *Zea mays* L.; (**b**) class mh1: *Lolium perenne*; (**c**) class mh2: *Sonchus oleraceus*; (**d**) class mh3: *Solanum nigrum*; (**e**) class mh4: *Poa annua*.

### 2.2.3. Training Configuration Parameters

The Yolov5s v7.0 object detection algorithm developed by [32] was used. This algorithm is an open-source real-time image object detection system that uses a single convolutional neural network (CNN) pre-trained with the COCO dataset. The pre-trained weight model YOLOv5s.pt was selected because, in the work developed by [29], the YOLOv5s.pt model showed better performance over the other models of the Yolo-v5 family when used on the same crop. Additionally, in the work of [33], the author states that the models YOLO-v5n and YOLO-v5s showed advantages in inference times, being faster than YOLO-v4 but with lower accuracy, and concluded that all YOLO models, especially YOLO-v5n and YOLO-v5s, have shown great potential for real-time weed detection, and increased data could improve detection accuracy. Similarly, in the work of [20], the author evaluated the different parameters of the YOLO family algorithms; one of them is inference time, which for YOLO-v5, showed a time less than 20 ms for most of the datasets, being less than YOLO-v6 and YOLO-v7, which achieved detection around 30 ms for the same datasets. Since the overall objective of this research is the design and construction of a field cart capable of quickly detecting weeds in corn crops and performing precision weeding, the image processing time is a critical factor, especially the inference time of a new image when it enters the model, so we chose YOLO-v5s to perform this experiment.

The file custommaiz1400.yaml was configured with the labels obtained from the open-source tool LabelImg. Three hundred (300) epochs were used for the training. This value was selected because, in previous training, the best weight of the network was reached in half of the epochs. In the work of [34], the authors trained the algorithm with different amounts of epochs, obtaining the highest accuracy with 600 epochs; thus, the previous training mentioned above was performed. The batch size was set to 32, a value that allows us to reduce the over-fitting of the model in the initial stages. As for image size, the YOLO-v5 models uniformly reduce the input images to 640 × 640 pixels. The initial (lr0) and final (lrf) learning rate hyperparameters were set to 0.01, the weight decay parameter was set to 0.0005, and the momentum was set to 0.937, which are values recommended by [32]. The other hyperparameters used have been those defined by the default model. The results of the between-performance of the algorithm were organized into a classification confusion matrix. The CNN performance was based on three metrics using the equations of [35]: Recall, F1-score, and mAP (mean Average Precision).

### 2.2.4. Equipment Configuration

The hardware and software configuration described in Table 1 was used to train and validate the model.

**Table 1.** Configuration used for model training.

| System Component | Description |
|---|---|
| Processor | Core(TM) i7-12650H 12th Gen Intel(R) 2.70 GHz |
| GPU | NVIDIA GeForce RTX 3050 4096MiB |
| RAM | 16 GB |
| Operating System | Windows 11 Enterprise |
| Accelerated environment | CUDA 11.7 |
| | Jupyter Notebook Versión 6.5.4 |
| Development environment | Python 3.8.16 |
| | PyTorch 2.0.0—Torchvision 0.15.0—Torchaudio 2.0. |

## 3. Results and Discussion

### 3.1. Training Results

As a result of the training, the best.pt file containing the weights of the model is obtained. From these values, the validation is performed with 350 images, and the following metrics are obtained as indicators of model performance.

3.1.1. Confusion Matrix

According to the results of the confusion matrix (Figure 3), the model demonstrated exceptional performance in locating and classifying corn and weeds. The highest prediction accuracy was observed for corn, at 97%. A high prediction was also obtained for mh2 (*Sonchus oleraceus*), at 90%. However, lower prediction accuracy rates were recorded for mh1 (*Lolium perenne*), 86%, mh3 (*Solanum nigrum*), 78%, and mh4 (*Poa annua*), 74%.

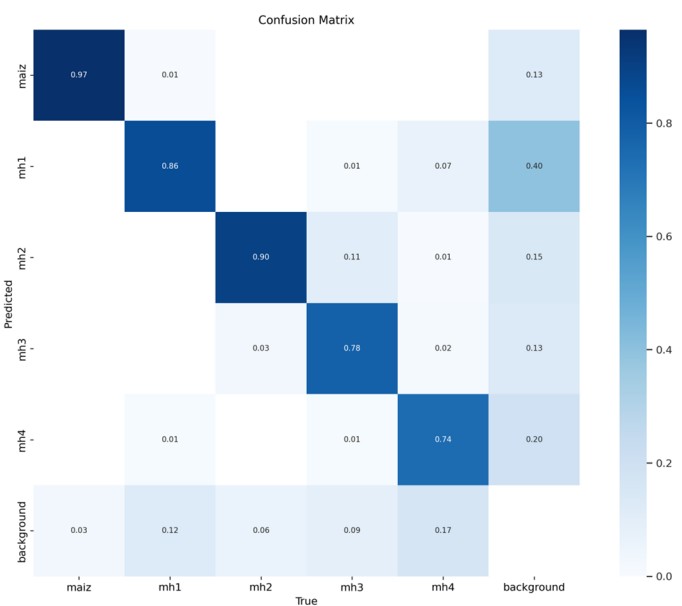

**Figure 3.** Confusion matrix for the 5 classes of the model.

Comparing the results of the confusion matrix with those of [29], which used the same algorithm and obtained a prediction of 92% for corn and 86% for weeds, it is possible that the prediction values of [29] are lower because the images were taken under different lighting conditions, which directly affects the prediction of the model. It is important to mention that among the weeds, the model is usually confused, specifically mh2 (*Sonchus oleraceus*) with mh3 (*Solanum nigrum*) in 11%. This is because in their early stages of development, the leaves usually have the same shape, and as they grow, the number of leaves and shape varies. The same happens with mh1 (*Lolium perenne*) and mh4 (*Poa annua*), with 7% corresponding to narrow leaves.

### 3.1.2. Precision–Recall Curve and F1-Score–Confidence Curve

For each epoch, the algorithm estimates the precision and recall and plots them to know the performance of the model (Figure 4), calculating the areas under the curves of each class and obtaining the mAP@05 values for each class.

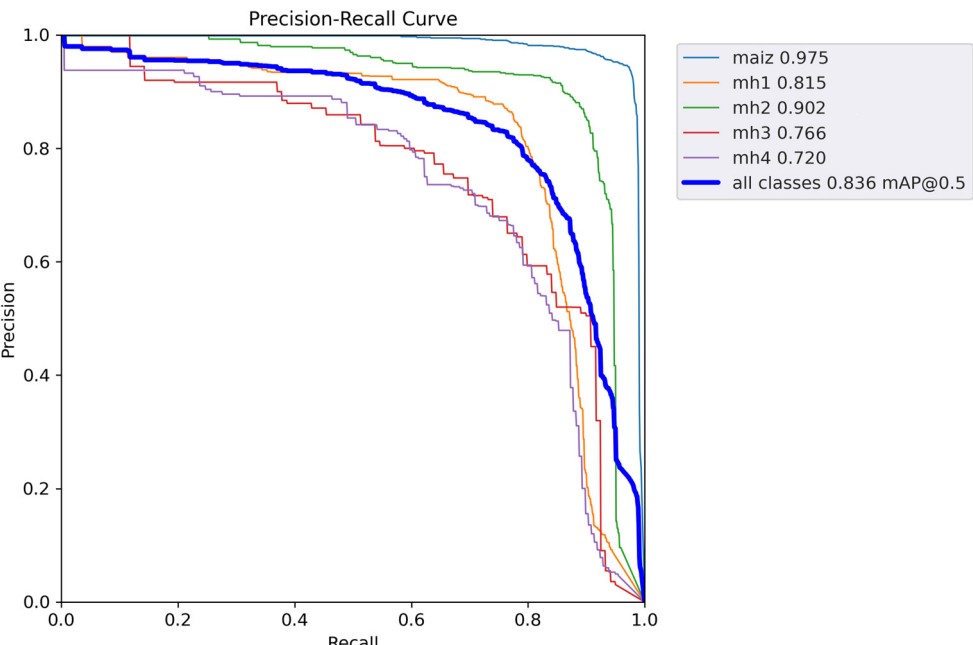

**Figure 4.** Precision–recall curve for the 5 classes.

mAP@05 calculates the average of the accuracies of each class at a confidence threshold of 0.5. The higher the value of mAP@05, the better the model performance. The maize class has the highest value of 0.975 mAP (97.5%), so its predictive power is higher than the other classes. The weed classes vary between 0.902 (90.2%) and 0.720 (72.0) mAP, respectively. For the model in general, the mean of the mAP of the five classes was calculated, obtaining a value of 0.836 (83.6%) mAP@0.5. The overall mAP value of the model (83.6%) is lower than that of the corn class (97.5%) because it is an average of the precision values of the other classes (weeds), making the detection efficiency of the model generally lower. However, this does not mean that its application in the field is inappropriate since if the aim is to discriminate corn from weeds regardless of the species, the model is suitable for implementation in a weeding system that seeks to eliminate weeds regardless of their species. In the work of [29], the values of mAP for corn were 96.3%, and those of weeds were 88.9%, values similar to those obtained by our model.

Similarly, for the precision–recall curve, the F1-score is calculated as a performance metric and plotted against the confidence scores, obtaining a mean value of 0.81 for all classes, as shown in Figure 5. Corroborating the above, the behavior is similar to that in Figure 3, showing that the corn class has the highest predictive power.

### 3.1.3. Loss Functions

Figure 6 presents the various loss functions and training evaluation metrics. The curves illustrate three types of loss functions: box loss (box_loss), object loss (obj_loss), and classification loss (clc_loss). Box loss represents the predictive coverage of the target object's bounding box, assessing the probability that a specific object is found in a region of interest. Object loss indicates whether an object is present in a given image window. On the other hand, classification loss refers to the discrepancy between the actual class of a predicted object and the expected class.

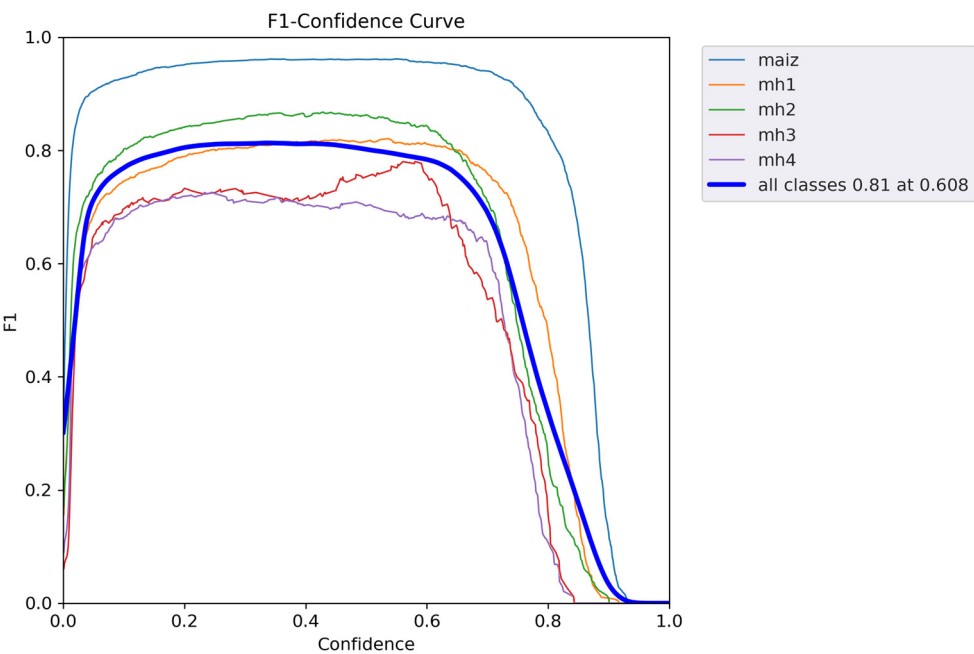

**Figure 5.** F1–confidence curve for the 5 classes.

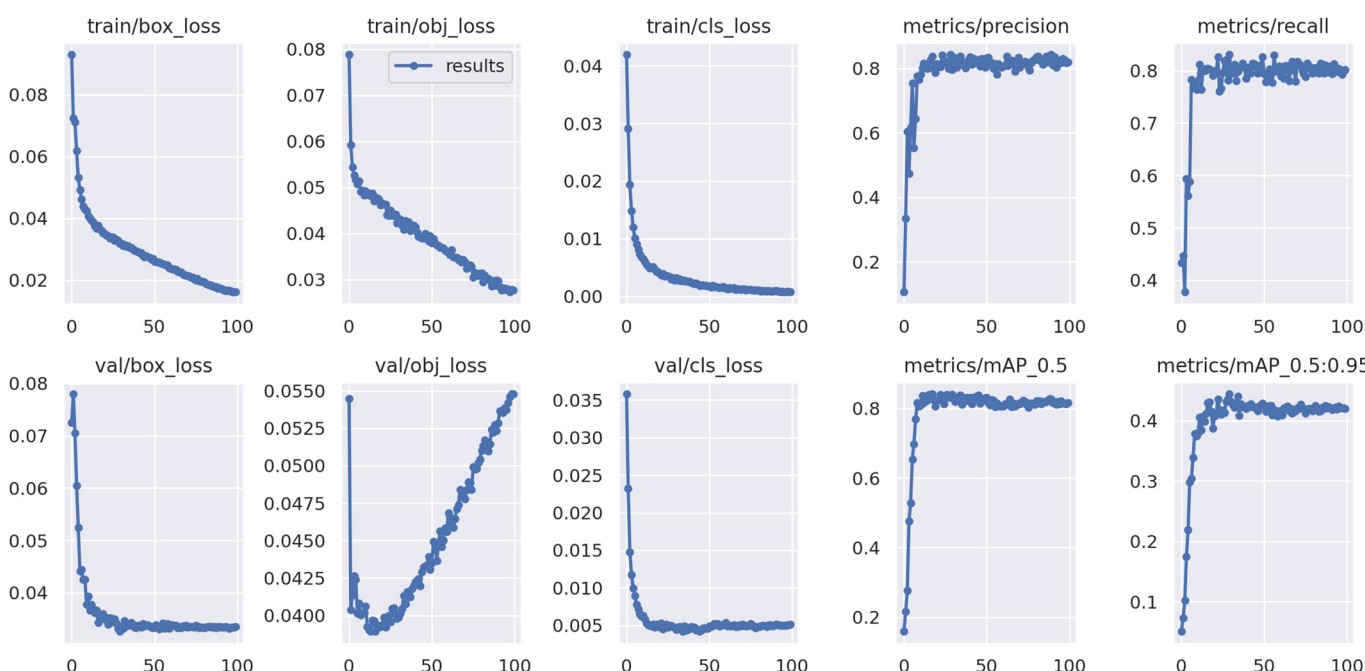

**Figure 6.** Loss functions and performance metrics.

In Figure 6, it is clear that the performance of the model improves as training progresses. In addition, there is a progressive increase in accuracy and recall, both in the training set and in the tests, suggesting that the model does not suffer from overfitting. It is important to highlight that out of the 300 epochs configured, the algorithm performed only 143 since, from epoch 43, the model reached convergence. It can be observed how the model seems to converge even before epoch 43 is mentioned in the validation. On the other hand, in the val/obj_loss graph, an increase in loss is observed after the first 10–20 epochs. The same is true for precision and mAP plots. The values show a stabilization and even a decrease after these 30–40 epochs.

### 3.1.4. Verification of Results

Additional verification of the results of the performance metrics was performed to visually evaluate the results of the model. For this, a set of images from different stages of crop development were selected, instantiated, and compared with the images before labeling. The values in the bounding boxes correspond to the following classes: 0: maize, in blue; 1: mh1(*Lolium perenne*), in green; 2: mh2 (*Sonchus oleraceus*), in red; 3: mh3 (*Solanum nigrum*), in cyan; and 4: mh4 (*Poa annua*) in yellow. Figure 7a,b shows the crop 15 days after planting, the corn plants identified, and the weeds that emerged at the date that the model found and correctly classified.

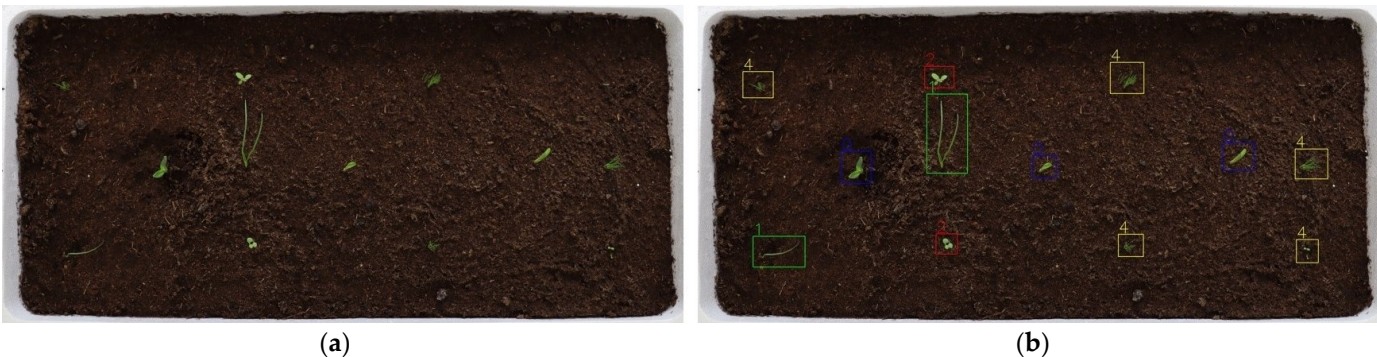

(**a**)　　　　　　　　　　　　　　　　　　(**b**)

**Figure 7.** (**a**) unprocessed image of the crop 15 days after planting; (**b**) processed image of the crop 15 days after planting. The values in the bounding boxes correspond to the following classes: 0: maize, in blue; 1: mh1(*Lolium perenne*), in green; 2: mh2 (*Sonchus oleraceus*), in red; and 4: mh4 (*Poa annua*) in yellow.

Figure 8a,b shows the crop 30 days after planting, in which the processed image identified the corn plants correctly; however, within a bounding box of the third corn plant, mh4 (4-yellow) is hidden, and in Figure 7b, if it was properly separated, this could be corrected by increasing the number of images with these characteristics during training. One of the possible causes of maize being grouped with the weed class mh4 (*Poa annua*) is its shape; it has narrow, thin leaves that can be similar during vegetative development approximately between days 20 and 30. However, in Figure 9, it can be observed that the mh4 is located under one of the primary leaves of maize, so the algorithm can interpret it as if it were a single object; this could be corrected by increasing the number of images with this behavior in the training and thus have greater accuracy when this condition occurs.

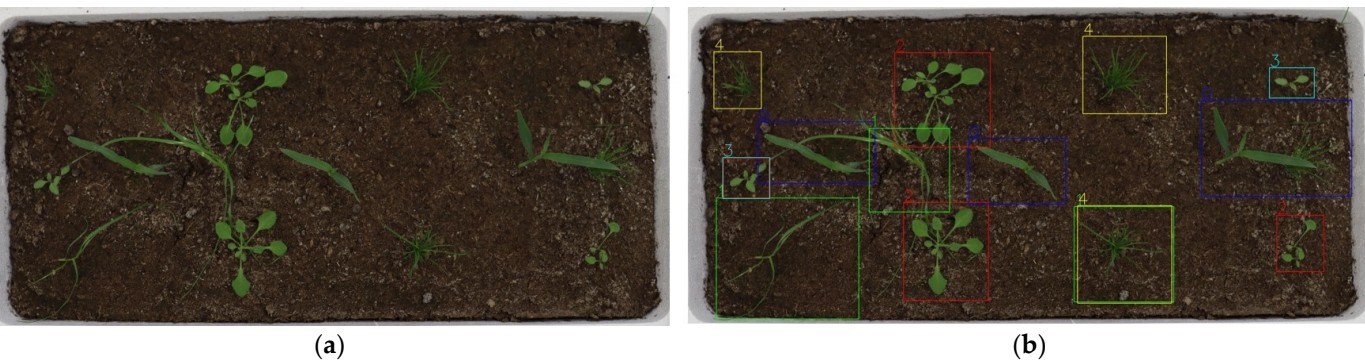

(**a**)　　　　　　　　　　　　　　　　　　(**b**)

**Figure 8.** (**a**) unprocessed image of the crop 30 days after sowing; (**b**) processed image of the crop 30 days after sowing. The values in the bounding boxes correspond to the following classes: 0: maize, in blue; 1: mh1(*Lolium perenne*), in green; 2: mh2 (*Sonchus oleraceus*), in red; 3: mh3 (*Solanum nigrum*), in cyan; and 4: mh4 (*Poa annua*) in yellow.

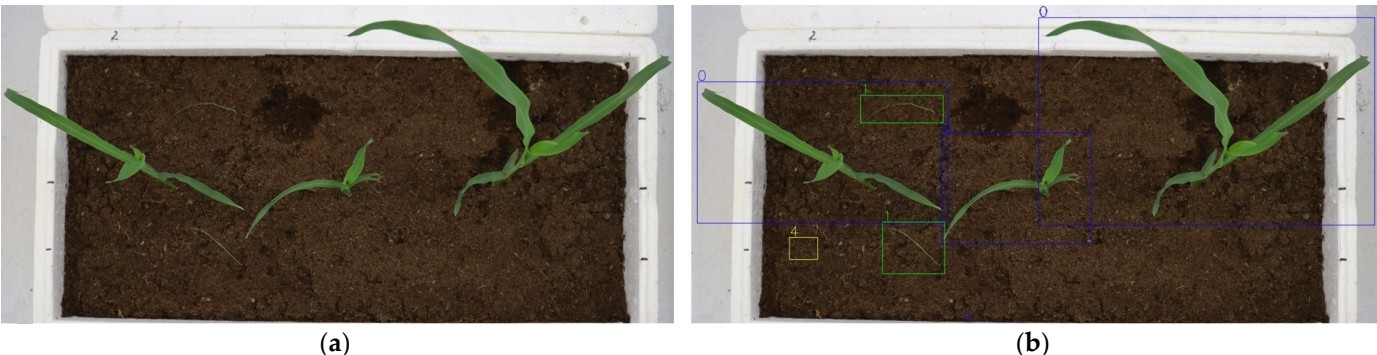

**Figure 9.** (**a**) raw image of the crop 45 days after sowing; (**b**) processed image of the crop 45 days after sowing. The values in the bounding boxes correspond to the following classes: 0: maize, in blue; 1: mh1(*Lolium perenne*), in green; and 4: mh4 (*Poa annua*) in yellow.

As for the weeds, in class mh1 (1-green), of the two plants present, the model identified three, i.e., it classified one more, presenting a false positive case, identifying it as mh4 (4-yellow). The similarity between the leaves (narrow) of the two classes means that incorrect identification is possible. For the mh4 (4-yellow) class, of the four (4) plants present, two (2) were correctly classified, and the third was correctly classified, but the system also classified it as mh1 (1-green), co-proving the above-mentioned case. The system usually classifies these weeds as belonging to the two classes. The fourth plant mh4 (4-yellow), was not identified by the model, and as it was not classified, it was identified in the bounding box of the third corn. In mh2 (2-red), the system found four (4) of the three (3) planted, classifying one more, producing another false positive, a similar occurrence to the two previous classes. The shape of the leaf, in this case, is wide, and the system can classify it into two categories. As for the mh3 (3-cyan), of the two (2) present, the system classified them correctly; only one (1) was classified twice. Figure 9a,b shows the crop 45 days after sowing, and the corn plants are correctly classified. The new weeds emerging from the second planting were classified correctly.

In [36], the authors built a CNN based on the light model YOLOv4-tiny called YOLOv4-weeds to detect weeds in the maize crop; the results were compared with five different CNNs: Faster R-CNN, SSD 300, YOLO-v3, YOLOv3-tiny, and YOLOv4-tiny, the comparison was made with the average mean accuracy (mAP). The proposed YOLOv4-weed model obtained the best value with 86.69%, while the Faster R-CNN model obtained 68.55%, SSD300 achieved 69.22%, YOLO-v3 achieved 85.55%, YOLOv3-tiny achieved 75.72%, and YOLOv4-tiny achieved 80.33%. The mAP value obtained in our research for YOLO-v5s was 97.5% for the maize class, which is higher than the mAP obtained in [36]; this is mainly due to the variation in the illumination of the field conditions. In [28], the author evaluated different versions of the YOLO family, specifically YOLO-v4, YOLOv4-tiny, YOLOv4-tiny-3l, and YOLO-v5 versions s, m, and l, to detect and count corn plants in a field with the presence of weeds, and the results obtained in terms of mAP show that the best score was for YOLO-v5s, with 73. 1%; the other models obtained the following values: YOLO-v4, 72.01%; YOLOv4-tiny, 64.91; YOLOv4-tiny-3l, 58.06%; YOLO-v5m, 71.6%; and YOLO-v5l, 68.53%. In this case, the images were taken with a UAV, which implies that the resolution is lower than our study but confirms that the best version was YOLO-v5s.

In [37], five previously trained last-generation CNNs, ResNet50, VGG16, VGG19, Xception, and MobileNetV2, were evaluated to classify weeds and crops into their respective classes among corn crops. The accuracy results are as follows: ResNet50, 95.23%; VGG16, 93.32%; VGG19, 93.32%; Xception, 93.59%; and MobileNetV2, 93.5%. ResNet50 yielded the best results in terms of accuracy, approaching 97.5% of YOLO-v5s; it is important to mention that a set of images of different species, both weeds and cash crops, were evaluated; it is possible to achieve better accuracy when evaluating the crops separately. In Wang's work, he proposed a lightweight model based on YOLOv5s to build a precision spraying

robot in maize crops. They used YOLO-v5s, YOLO-v5l, YOLO-v5m, and YOLO-v5x versions, comparing their accuracy. They found that the best version was YOLO-v5s, with a mAP of 93.2% for corn and a mAP of 85.9% for weeds. After finding the best version, they modified its structure by adding a C3-Ghost-bottleneck module, with which they improved YOLO-v5s, obtaining a mAP value for corn of 96.3% and a mAP for weeds of 88.9%. The results found by [29] are close to those obtained in our study, and he claims that the YOLO-v5s model can be integrated into automatic precision weeding systems. Similarly, [30] designed a weeding robot based on the CNN YOLOX to remove weeds in corn seedling fields to verify the feasibility of using a blue light laser as a non-contact weeding tool. The detection accuracy of the model was 92.45% for corn and 88.94% for weeds.

## 4. Conclusions

A mobile support structure for image acquisition was built and used to train the YOLOv5s model. This model implemented a CNN for the classification of five classes, obtaining a prediction of 97% and a mAP@05 of 97.5% for corn. For weeds, we obtained the following: mh1(*Lolium perenne*) with 86% prediction and a mAP@05 of 81.5%; mh2 (*Sonchus oleraceus*) with 86% prediction and a mAP@05 of 90.2%; mh3 (*Solanum nigrum*) with 78% prediction and a mAP@05 of 76.6%; and mh4 (*Poa annua*), with 74% prediction and a mAP@05 of 72.0%. The overall model for all classes had a mAP@05 of 83.6%. The prediction and the mAP@05 of the corn class are the highest of all the classes of the model. This is an encouraging result for the project in general, which seeks to build a precision weeding system, since if the algorithm identifies the plant to protect (corn) with high probability and differentiates itself from the weeds, the system will seek to eliminate the weed regardless of its accuracy. However, it is necessary to continue feeding the model with images from new campaigns and enrich the set of training images. The application of CNN models integrated with artificial vision systems opens a wide range of application possibilities, not only what is proposed in this work, which is to perform precision weeding, creating a system capable of eliminating weeds with high precision in a selective manner, but also to integrate algorithms that at the same time enable monitoring the state of the crop, having a constant record of growth. In addition, this technology also allows the monitoring of pest or disease affectations, making it possible to make timely decisions about preventive or corrective measures. These new algorithms that can be integrated into current weeding systems would have to be built and trained for each crop, which opens a window of future work for different researchers. As for precision weeding, CNN models could be integrated with other technologies to eliminate weeds without agrochemical application; for example, it could be integrated with pressurized water jet systems that cut the weeds, and it could also be integrated with laser weed burning systems, allowing for clean agriculture. This approach would contribute to advancing technology applied to agriculture for more efficient and sustainable crop management.

Future work: Within this project plan, several phases were contemplated to create a robust system for the detection of weeds in maize crops; the first phase was to create an image dataset under controlled conditions in a greenhouse and create the first CNN model for real-time detection, as presented in this work. The subsequent phases are to acquire images in adverse field conditions and to use a learning transfer model to strengthen the current model, for which a new growing campaign will be conducted to feed the set of training images to improve the classification in the cases found as anomalous, wherein the new results will be validated in the field, and a precision weeding system integrated to the mobile structure presented in this work will be built. Additionally, the evaluation of other last-generation CNNs will be extended to obtain a higher accuracy and speed of inference, with the objective of integrating it into the new precision weeding system.

**Author Contributions:** Conceptualization, O.L.G.-N. and L.M.N.-G.; methodology, O.L.G.-N., O.S., P.M.-R. and M.Á.V.-M.; software, O.L.G.-N. and P.M.-R.; validation, O.L.G.-N., O.S. and L.M.N.-G.; formal analysis, O.L.G.-N., P.M.-R., M.Á.V.-M. and L.M.N.-G.; investigation, O.L.G.-N., O.S. and L.M.N.-G.; resources, P.M.-R. and L.M.N.-G.; writing—original draft preparation, O.L.G.-N. and

L.M.N.-G.; writing—review and editing, O.L.G.-N., M.Á.V.-M. and L.M.N.-G.; supervision, O.L.G.-N., P.M.-R. and L.M.N.-G.; project administration, L.M.N.-G.; funding acquisition, L.M.N.-G. All authors have read and agreed to the published version of the manuscript.

**Funding:** This research was funded by the European Union through the Horizon Europe Program (HORIZON-CL6-2022-FARM2FORK-01) under project 'Agro-ecological strategies for resilient farming in West Africa (CIRAWA)'. Oscar Leonardo García-Navarrete was financed under the call for University of Valladolid predoctoral contracts, co-financed by Banco Santander.

**Institutional Review Board Statement:** Not applicable.

**Data Availability Statement:** The data presented in this study are available upon request from the corresponding author.

**Acknowledgments:** The authors would like to thank Huercasa, S.A. for supplying the seed and its cultivation recommendations and the greenhouse staff of the Department of Crop Production and Forestry Resources of the University of Valladolid for their help during the experiments.

**Conflicts of Interest:** The authors declare no conflicts of interest.

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
