# Peer review of "Development of a Detection System for Types of Weeds in Maize (Zea mays L.) under Greenhouse Conditions Using the YOLOv5 v7.0 Model"

_agriculture, doi:10.3390/agriculture14020286_

Round 1
Reviewer 1 Report
Comments and Suggestions for Authors
1. Part 2.1.1,Please provide information on time, temperature, humidity, growth period, etc.
2.Part2.1.1和Part2.2.1, confusing.
3.Please provide images for five categories.
4.Please increase the discussion on which grass is easily confused with corn seedlings at which stage.
5.Why only YOLOv5 was used instead of YOLOv8? It is recommended to compare other deep learning algorithms and increase the impact factor analysis.
Comments on the Quality of English Language
Minor edit
Reviewer 2 Report
Comments and Suggestions for Authors
The manuscript titled "Development of a detection system for types of weeds in maize (Zea mays L.) under greenhouse conditions using the YOLOv5 v7.0 model" primarily addresses the development of an artificial vision system for differentiating maize from four types of weeds. Its innovative approach meets specific needs in precision agriculture and weed management.
The research adds to the field by implementing a YOLOv5-based model to accurately identify weeds, a significant contribution compared to other published materials. The article thoroughly describes the experimental design and methods, including materials used, image acquisition system, data annotation, training configuration, and model performance evaluation.
The article's conclusions are in line with the presented evidence, offering a robust solution for weed removal. The references are appropriate and well-cited, supporting the research's context and findings effectively. The language and grammar of the article are professional and appropriate for academic publication.
However, there are some issues to note:
1. In line 64, the authors mention that the latest version in the YOLO series is YOLO-v8, and in line 99, it is stated that the literature indicates YOLO-v8 has the best performance in weed detection. However, it is unclear why the authors chose the older YOLO-v5 model over the more recent YOLO-v8 for detecting maize and weeds.
2. There are some spelling errors in the text, such as "yo-lov5s.pt" on line 186, which should be "yolo-v5s.pt," and "algo-rhythm" on line 192, which should be "algorithm."
3. The authors have not made any modifications to the model but have merely utilized it. There is also no comparison with any related models, only an analysis of the YOLOv5s model's experimental results. It is recommended that the authors include comparisons with other relevant object detection models for more convincing results, such as the Two-stage Faster R-CNN series and transformer-based DETR series models.
4. The authors have not provided a link to the dataset used in their study. It is recommended that they include these links in their manuscript to foster further research and development in this field.
Despite these issues, the article is a strong candidate for publication, subject to the suggested major revisions and enhancements.
Comments on the Quality of English Language
The language and grammar of the article are professional and appropriate for academic publication.
Reviewer 3 Report
Comments and Suggestions for Authors
I find this an interesting and timely manuscript describing an innovative method for weed detection in maize using YOLO.
I have some suggestions to improve the manuscript:
1) I would reduce the introduction, it is too long
2) I would clearly justify the choice of a protected environment for the trial
3) I would separate results from discussion increasing the comparison with other studies
4) In the conclusions I would add some possible scenarios for the final application of this technology in farm machinery.
Good luck for the publication of this manuscript.
Round 2
Reviewer 2 Report
Comments and Suggestions for Authors
accepted as is